# From Parse-Execute to Parse-Execute-Refine: Improving Semantic Parser for Complex Question Answering over Knowledge Base

**Wangzhen Guo    Linyin Luo    Hanjiang Lai**[*]    **Jian Yin**
School of Computer Science and Engineering, Sun Yat-Sen University
{guowzh6, luoly36}@mail2.sysu.edu.cn
{laihanj3, issjyin}@mail.sysu.edu.cn

## Abstract

Parsing questions into executable logical forms has showed impressive results for knowledge-base question answering (KBQA). However, complex KBQA is a more challenging task that requires to perform complex multi-step reasoning. Recently, a new semantic parser called KoPL (Cao et al., 2022a) has been proposed to explicitly model the reasoning processes, which achieved the state-of-the-art on complex KBQA. In this paper, we further explore how to unlock the reasoning ability of semantic parsers by a simple proposed parse-execute-refine paradigm. We refine and improve the KoPL parser by demonstrating the executed intermediate reasoning steps to the KBQA model. We show that such simple strategy can significantly improve the ability of complex reasoning. Specifically, we propose three components: a parsing stage, an execution stage and a refinement stage, to enhance the ability of complex reasoning. The parser uses the KoPL to generate the transparent logical forms. Then, the execution stage aligns and executes the logical forms over knowledge base to obtain intermediate reasoning processes. Finally, the intermediate step-by-step reasoning processes are demonstrated to the KBQA model in the refinement stage. With the explicit reasoning processes, it is much easier to answer the complex questions. Experiments on benchmark dataset shows that the proposed PER-KBQA performs significantly better than the stage-of-the-art baselines on the complex KBQA.

## 1   Introduction

Knowledge-base question answering (KBQA) (Lan et al., 2022) is a task that answers natural language questions over a given knowledge graph. KBQA has become a hot research topic since it can lead to a large number of applications, e.g., intelligent assistants.

Many algorithms (Zhang et al., 2018; Yih et al., 2016; Schlichtkrull et al., 2018; Das et al., 2021; Cao et al., 2022a) have been proposed for KBQA. A notable research line for KBQA is semantic parsing (Yih et al., 2016; Cao et al., 2022a). It encodes the questions into formal meaning representations/logical forms (Lan et al., 2022), then the logical forms are executed over the knowledge base to give the final answers, which is called parse-execute paradigm. KaFSP (Li and Xiong, 2022) is a knowledge-aware fuzzy semantic parsing framework for KBQA. RnG-KBQA (Ye et al., 2021) is a rank-and-generate approach for solving generalization problem in KBQA. Several benchmarks KBQA datasets have also been proposed to promote research in this field. For example, MetaQA (Zhang et al., 2018), WebQSP (Yih et al., 2016) and LC-QuAD (Dubey et al., 2019) are the datasets for KBQA.

Recently, due to the complexities of users' information requirements, the more challenging task called *complex KBQA* (Lan et al., 2022) has been studied, which requires multi-step/compositional reasoning. For complex KBQA, it is important that the QA models can be learnt to explicitly model the reasoning processes, which can provide better interpretation of compositional reasoning. Hence, Cao et al. (2022a) had proposed a benchmark dataset KQA Pro and a knowledge-oriented programming language (KoPL). KQA Pro is a large-scale dataset for complex KBQA and provides explicit reasoing process for each QA pair. KoPL (Cao et al., 2022a) is a compositional and interpretable programming language for complex reasoning. The KoPL parser provides human understandable descriptions of logical forms. Based on the large-scale pretrained language model BART (Lewis et al., 2019), the BART-based KoPL has shown impressive results on KQA Pro dataset.

Although KoPL can explicitly describe the reasoning processes, the logical forms generated from

---

*Corresponding Author

semantic parser are the intermediate results but not the final answer, which may result in sub-optimized solutions. For example, the slight generation deviation (Das et al., 2021) may cause the logical forms to run incorrectly. Thus a natural question arises: *Can we use these intermediate results to further improve the complex reasoning ability?*

In this paper, we further explore how to unlock KoPL and KBQA model for improving its capability of complex reasoning. We find that the complex reasoning ability can benefit from demonstrating the KoPL's intermediate results to the KBQA model. That is, with the intermediate step-by-step reasoning processes from KoPL, we can elicit the multi-step reasoning for complex KBQA and achieve better performances compared to KoPL. Observed from that, we propose a novel parse-execute-refine paradigm for complex KBQA.

Figure 1 shows an overview of the proposed approach called PER-KBQA, which includes a parsing stage, an execution stage, and a refinement stage for complex KBQA. Given a question, the parser first utilizes a sequence-to-sequence model to generate the logical forms. Since KoPL can explicitly describe the reasoning process, we use KoPL as the logical forms in the parsing stage. Then, the execution stage aligns and executes the generated logical forms over the knowledge base. With that, we can obtain the intermediate step-by-step reasoning processes. Finally, in the refinement stage, the intermediate reasoning results are used as execution contexts. We combine the question and the execution contexts as inputs to the KBQA model. The main idea of this paradigm is that the answer may benefit from the intermediate step-by-step reasoning processes.

We evaluate the PER-KBQA on KQA Pro dataset and the extensive experimental results show that 1) the proposed PER-KBQA performs significantly better than the existing state-of-the-art baselines, and 2) PER-KBQA is a simple method that can further unlock the KBQA model for improving its complex reasoning ability. Besides, PER-KBQA provides insight into a novel way to combine the intermediate results into the final answer.

## 2 Related Work

**KBQA.** Complex Knowledge Base Question Answering aims to answer complex questions over knowledge bases, which always involves multi-hop reasoning, constrained relations and numer-ical operations (Lan et al., 2022). The existing solutions for Complex KBQA can be divided into two mainstream approaches, known as **semantic parsing-based (SP-based)** methods and **information retrieval-based (IR-based)** methods.

SP-based methods (Yih et al., 2016; Das et al., 2021) follow a parse-then-execute paradigm: parse a question into a logical form such as SPARQL or KoPL, and execute it against the KB to obtain the answer. IR-based methods (Miller et al., 2016; Saxena et al., 2020) follow a retrieve-and-generate paradigm: retrieve a question-specific graph and directly generate answer with text encoder-decoder. SP-based methods rely heavily on the parser to generate a logical form for each question, while they enjoy the advantage of powerful sequence-to-sequence pretrained language models. IR-based methods apply the answer generation module to make accurate prediction conditioned on the question and the retrieved context. IR-based methods require the retrieved context to be helpful.

Recently, the KoPL parser is proposed from KQA Pro (Cao et al., 2022a), and it is transparent about the execution process and can provide luxuriant information of the intermediate process, which makes it possible for our proposed method to combine both of the mainstream methods to make full use of their strengths.

**ODQA.** Open domain question answering aims to answer factoid questions based on a large collection of documents. The mainstream solution for open domain QA is that firstly use a retriever to select a small subset of passages where some of them might contain the answer to the question, and then utilize an extractive or generative reader to identify the answer. The retriever can be sparse vector space models such as TF-IDF (Chen et al., 2017) and BM25 (Robertson et al., 2009), or trainable dense vector space models such as DPR (Karpukhin et al., 2020) and MDR (Xiong et al., 2020). The extractive reader predicts a span from the retrieved context to answer an open domain question in previous works such as DrQA (Chen et al., 2017), HGN (Fang et al., 2019) and DPR (Karpukhin et al., 2020). Recently, the generative reader achieved further improvements in multiple open domain QA benchmarks, such as FiD (Izacard and Grave, 2020), KG-FiD (Yu et al., 2021) and PATH-FiD (Yavuz et al., 2022). In our work, we can view the proposed refinement stage as the generative reader, and the proposed parsing and execution

**Question 1:** What country is associated with Quincy, known as Gem City, where Mary Astor was born?

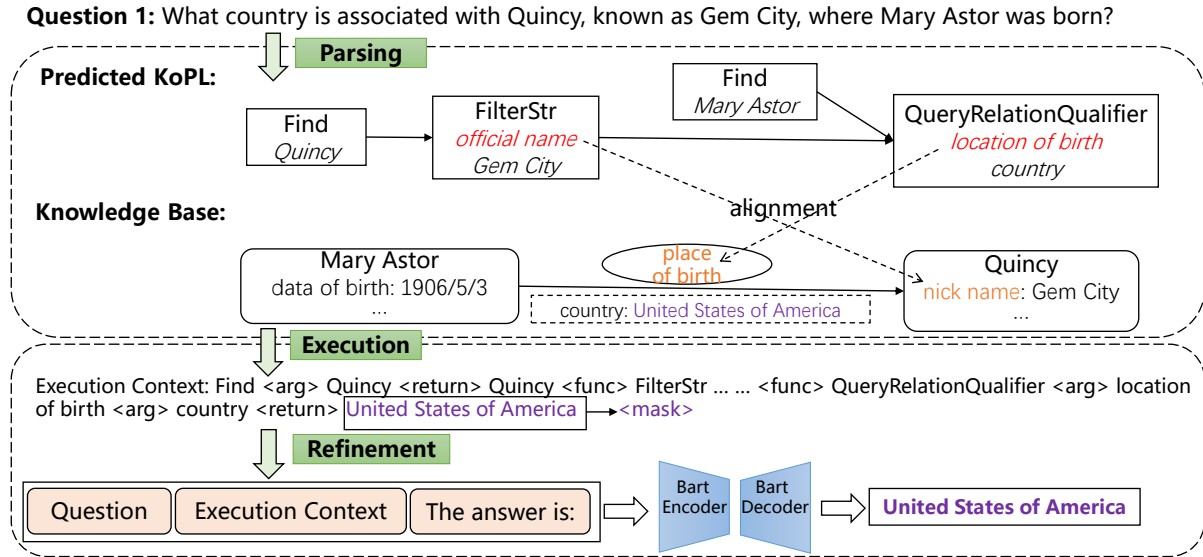

Figure 1: An overview of PER-KBQA approach. It consists of three stages: Parsing, Execution and Refinement. Parsing stage generated the logical form based on the question, and then execution stage aligned the knowledge base and obtained the execution context, and finally refinement stage generated the answer conditioned on the question and the execution context.

stages as the retrieval in ODQA.

## 3 PER-KBQA

In this section, we formulate the complex KBQA task and describe the implementation of three stages of PER-KBQA. In this paper, we consider the KoPL language that explicitly describes the reasoning process as our logical form. Since only the KQA Pro (Cao et al., 2022a) dataset provides the KoPL, we follow the KQA Pro setting. That is, each natural language question is paired with executable logical forms (i.e., KoPL). The logical forms can be executed against the KB to obtain the answer.

**Task Definition:** Let $q$ be a natural language query, $\mathcal{K}$ be a symbolic knowledge base and the training dataset be $\mathcal{D} = \{(q_1, l_1, A_1), (q_2, l_2, A_2), ..., (q_N, l_N, A_N)\}$, where $l_i$ represents the KoPL that can be executed against the $\mathcal{K}$ to obtain the ground truth answer $A_i$ corresponding to the query $q_i$. Given the knowledge base $\mathcal{K}$ and a question $q$, the semantic-parsing based QA methods are to generate the KoPL $\bar{l}$ and execute it to predict the answer $\bar{\mathcal{A}}$. In this paper, we explore how to use these logical forms that describes the reasoning process from KoPL to further unlock the KBQA model.

### 3.1 Parsing

The parsing stage models the logical form (i.e., KoPL) in a structured way. A sequence-to-sequence model is widely utilized to generate the logical form token by token conditioned on the question. Since the pretrained encoder-decoder language models (e.g., BART (Lewis et al., 2019)) have achieved considerable success on the question answering task, we leverage BART to generate KoPL following KQA Pro (Cao et al., 2022a).

Given the KoPL $l_i$, we convert its functions and arguments into the structured textual sequence:

$$l_i := P_1 <func> P_2 ... <func> P_n,$$
$$P_k := f_k(arg_{k1} <arg> arg_{k2}),$$

where $l_i$ consists of $n$ programs and the $k$-th program $P_k$ includes a function $f_k$ with its arguments $arg_{k1/k2}$. The function $f_k$ can have 27 forms as described in KQA Pro (Cao et al., 2022a). We insert special tokens $<func>, <arg>$ as indicators to separate the functions and their corresponding arguments respectively.

We train BART to autoregressively generate the $l_i$ per token at each step conditioned on the question $q_i$. Thus the conditional generation is to minimize cross-entropy loss as

$$\mathcal{L}(\theta^G) = -\frac{1}{N} \sum_{i=1}^{N} \sum_{j=1}^{|l_i|} \log p_{\theta^{\mathbf{P}}}(y_j | y_{<j}, q_i), \quad (1)$$

where $\theta^{\mathbf{P}}$ is the parameters of BART. The parsing stage is basically similar to the BART-based KoPL method proposed by KQA Pro (Cao et al., 2022a). Only the question is used to generate the logical form and the information in the knowledge base corresponding to the question is not utilized.

## 3.2 Execution

In the execution stage, we execute the generated logical forms over knowledge base to obtain the corresponding intermediate results. However, it is not guaranteed that the generated functional arguments will be consistent with the relations and attributes (edges) of the entities in knowledge base $\mathcal{K}$. For example, in Figure 1, the generated text *official name* and *location of birth* mismatch the attribute *nick name* and relation *place of birth* respectively. Although they are semantically similar, the logical forms can not be executed successfully in this knowledge base $\mathcal{K}$.

To alleviate this problem, we propose two steps: 1) Alignment. We first explicitly align the generated functional arguments $l_i$ with the relations and attributes in $\mathcal{K}$ based on their overlap; 2) Execution. We then execute the aligned logical form $l_i$ over the knowledge base $\mathcal{K}$, which obtains a list of entities as intermediate results for all programs in $l_i$.

More specially, in the alignment step, for each program $P_k$ in KoPL $l_i$, we define a candidate pool $\mathcal{P}_{P_k}$ that contains the entity-specific relations or attributes in the knowledge base. For example, in Figure 1, the candidate pool $\mathcal{P}_{FilterStr}$ contains the attributes' key-value pairs from the entity *Quincy*. For each generated functional arguments in $P_k$, we utilize the Jaccard similarity to find the most similar relations or attributes that exists in knowledge base $\mathcal{K}$:

$$arg_k^{aligned} = \arg\max_{a \in \mathcal{P}_{P_k}} \{ \frac{|a \cap arg_k|}{|a \cup arg_k|} \},$$

where $arg_k$ is the generated functional argument, $a \in \mathcal{P}_{P_k}$ is the relation or attribute in the knowledge base, and $arg_k^{aligned}$ is the aligned functional argument. Then the generated functional argument $arg_k$ is replaced by the most similar relation or attribute $arg_k^{aligned}$ in the knowledge base to make the logical form executable. If the generated argument exists in the knowledge base, it always aligns with itself. Otherwise, it aligns with the most similar edge in the knowledge base.

Previous work (Das et al., 2021) have studied the alignment methods such as using the cosine similarity of the words embeddings encoded by the pretrained language model BERT (Devlin et al., 2018) or encoded by the pretrained EmbedKG model TransE (Bordes et al., 2013). However, it is noted that we perform the alignment operation during the execution process and the candidate pool is always large, so we use the Jaccard similarity which is more efficient.

## 3.3 Refinement

After the parsing and execution stages, we obtain the execution contexts that consist of the logical forms and the corresponding intermediate results from the knowledge base, which describe the step-by-step reasoning process. The refinement stage utilizes the execution contexts to improve the complex reasoning ability.

Specifically, our proposed refinement stage organizes the predicted KoPL $\bar{l}_i$ and the intermediate results obtained from the execution stage in the form of the step-by-step reasoning path:

$$
\begin{aligned}
C_i &:= \bar{P}_1 <func> \bar{P}_2 \ ... <func> \bar{P}_n, \\
\bar{P}_k &:= f_k <arg> arg_{k1} \ ... \ <return> r_k, \quad (2) \\
r_k &:= e_1 \mid e_2 \mid ... \mid e_m,
\end{aligned}
$$

where $C_i$ is the execution context of logical form $\bar{l}_i$ and $r_k$ denotes the intermediate result of the $k$-th function that consists of a list of returned entities $\{e_i\}$. And the $<return>$ is a special token inserted as an indicator of the returned results. In our experiments, $m$, which is the number of entities that the $k$-th function $\bar{P}_k$ returns, ranges from 0 to 5. If there are more than 5 entities returned, we randomly sample 5 entities to reduce the expensive quadratic computation complexity about the input length.

Then we concatenate the question $q_i$ and the execution context $C_i$ as following:

$$
\begin{aligned}
X_i := \ &Question : q_i \\
&Context : C_i \\
&The \ answer \ is : \quad ,
\end{aligned}
$$

and obtain the transformed training set $\{(X_i, A_i)\}$ based on the original training set.

The goal of the refinement stage is to train another BART model to autoregressively generate the answer $A_i$ conditioned on $X_i$. Thus the answer generation is to minimize cross-entropy loss over

the transformed training set $\{(X_i, A_i)\}$ as

$$\mathcal{L}(\theta^R) = -\frac{1}{N} \sum_{i=1}^{N} \sum_{j=1}^{|A_i|} log p_{\theta^{\mathbf{R}}}(y_j | y_{<j}, X_i), \quad (3)$$

where $\theta^{\mathbf{R}}$ is the parameters of the answer refinement model.

It is noted that the last function $\bar{P}_n$ returns the answer $r_n$ (may be correct or incorrect) when executing the predicted logical form $\bar{l}_i$. We mask the returned answer $r_n$ of the $\bar{P}_n$ in the context $C_i$ with a certain probability $p$ when training the answer generation model. So we can avoid learning the spurious correlation between the returned answer $r_n$ and the gold answer $A_i$, and make the model attend to the question and reasoning path to predict the answer. We further explore the mask strategy in the ablation study.

There are some motivations to introduce the refinement stage. 1) The logical forms are the intermediate outputs but not the final answer. Thus the logical forms may not be executed successfully against the knowledge base $\mathcal{K}$ to obtain a reasonable answer because of the slight generation deviation (Das et al., 2021), even though the alignment operation is performed. Fortunately, the intermediate reasoning processes from KoPL are transparent and explainable, which can provide abundant intermediate results as clue information to improve the capability of complex reasoning. 2) Recently, fusion-in-decoder (FID) (Izacard and Grave, 2020) and its extension work, e.g., KG-FiD (Yu et al., 2021) and PATHFiD (Yavuz et al., 2022), have shown impressive performances. These QA methods leverage the passage retrieval with generative reader for open domain QA, and have shown that the QA models can benefit from the context from the retrieval module. Inspired by that, our proposed method also takes advantage of the execution context for improving performances. In this paper, we use the intermediate step-by-step reasoning processes as the execution contexts instead of the passage retrieval. From this perspective, we can view the parsing and execution stages as retriever and consider the refinement stage as a generative reader in FID (Izacard and Grave, 2020). 3) Inspired by the chain-of-thoughts prompting methods (Wei et al., 2022; Kojima et al., 2022; Li et al., 2022), which have shown that the intermediate reasoning steps can improve the ability of pre-training language models to complex reasoning, we thus use the intermediate reasoning steps from KoPL

to further refine the language model for complex reasoning. This is also why we choose KoPL as the semantic parser since it can provide transparent and human understandable reasoning processes, which are more similar to natural languages.

## 4 Experiments

In this section, we conduct the experiments to compare our proposed approach with various baselines, and provide detailed analysis and ablation study on the results.

**Dataset:** Since KoPL is a recently proposed parser that explicitly describes the reasoning process and only the KQA Pro (Cao et al., 2022a) dataset provides KoPL, we use KQA Pro, a challenging dataset for complex question answering over knowledge base for evaluation.

In KQA Pro training set, each instance consists of the textual question, the corresponding KoPL, ten candidate choices, and the golden answer. And there are two settings: open-ended setting and multiple-choice setting. We conduct our experiments in the open-ended setting where the ten candidate choices are not used. Besides, it provides a high-quality knowledge base which has rich literal knowledge (attributes and relations about entities) as well as qualifier knowledge (qualifier about attributes and relations). We can execute KoPL over the knowledge base to obtain the answer.

**Metrics:** We evaluate the QA models based on the accuracy of the test set in KQA Pro dataset. Following the KQA Pro (Cao et al., 2022a), we provide in-depth analysis of models' fine-grained reasoning abilities for Complex KBQA including Multi-hop, Qualifier, Comparison, Logical, Count, Verify, and Zero-shot, where *Multi-hop* denotes multi-hop questions, *Qualifier* and *Comparison* measure the ability to query the literal knowledge or qualifier knowledge of the relations or attributes, *Count* and *Verify* rely on the intermediate results of entities list, *Zero-shot* requires the QA models to answer the questions related to unseen relations.

**Baselines and Implementation Details:** We reproduce the strong baseline BART based KoPL parser proposed in KQA Pro (Cao et al., 2022a). Based on this baseline, we implement three stages of our approach. We also report the published methods in the leaderboard including KVMNet (Miller et al., 2016), SRN (Qiu et al., 2020), EmbedKGQA (Saxena et al., 2020), RGCN (Schlichtkrull et al., 2018), GraphQ IR (Nie et al., 2022). Following

| Model | Overall | Multi-hop | Qualifier | Comparison | Logical | Count | Verify | Zero-shot |
|---|---|---|---|---|---|---|---|---|
| KVMemNet | 16.61 | 16.50 | 18.47 | 1.17 | 14.99 | 27.31 | 54.70 | 0.06 |
| SRN | - | 12.33 | - | - | - | - | - | - |
| EmbedKGQA | 28.36 | 26.41 | 25.20 | 11.93 | 23.95 | 32.88 | 61.05 | 0.06 |
| RGCN | 35.07 | 34.00 | 27.61 | 30.03 | 35.85 | 41.91 | 65.88 | 0.00 |
| BART SPARQL | 89.68 | 88.49 | 83.09 | 96.12 | 88.67 | 85.78 | 92.33 | 87.88 |
| BART KoPL | 90.55 | 89.46 | 84.76 | 95.51 | 89.30 | 86.68 | 93.30 | 89.59 |
| GraphQ IR | 91.70 | 90.38 | 84.90 | 97.15 | 92.64 | **89.39** | 94.20 | **94.20** |
| PER-KBQA(ours) | **93.82** | **92.93** | **89.83** | **97.57** | **92.99** | 89.01 | **95.30** | 91.69 |

Table 1: Accuracy of various baselines and our method on KQA Pro test set.

| PE | $q_i$ | $C_i$ | recall |
|---|---|---|---|
| ✓ | | | 93.26 |
| | ✓ | ✓ | 95.27 |

Table 2: The recall of the golden answer on KQA Pro validation set. PE denotes that we utilize Parsing and Execution stages to generate the answer. And $q_i, C_i$ represent the question and execution context respectively.

BART KoPL (Cao et al., 2022a), we utilize two independent bart-base models to instantiate the parsing and refinement stages respectively. We set the learning rate for BART parameters as 3e-5, the learning rate for other parameters as 1e-3, and the weight decay as 1e-5. We used the optimizer Adam (Kingma and Ba, 2014) for all models. Please note that our proposed execution stage is simple, efficient and requires no learning.

### 4.1 Quantitative Results

**Overall Results.** In Table 1, we compare the experimental results of our proposed method with the existing approaches for KQA Pro in the same open-ended setting. We observe that our method PER-KBQA outperforms the existing published work on the KQA Pro benchmark. Compared to the sota, our proposed method shows a significant performance gain in terms of most evaluated measures. In particular, our method improves the various complex reasoning skills for the QA system, e.g., multi-hop reasoning and qualifier knowledge query.

Compared to the sota method BART KoPL (Cao et al., 2022a) which follows parse-execute paradigm, our proposed parse-execute-refine paradigm performs significantly better and has several advantages. 1) First, we can explicitly align the knowledge base in the execution stage. Because of

the KoPL parser that provides the transparent reasoning steps, our execution stage can make alignment efficiently from the proposed candidate pool instead of a huge comparison space. With the proposed execution stage, we can successfully obtain the intermediate results instead of failing to execute the programs when there are slight generation deviations. 2) Second, the intermediate reasoning steps make it much easier to perform complex reasoning. The refinement stage uses the intermediate results to further generate a reasonable answer. Since the execution context can provide abundant information of the reasoning process including reasoning steps and their intermediate results, they are important clues to the answer. Therefore, our proposed parse-execute-refine paradigm is beneficial to all measures including Multi-hop, Qualifier, Comparison, Logical, Count, Verify, and Zero-shot.

Table 2 is an example that shows it is beneficial to demonstrate the executed intermediate reasoning steps to the KBQA model. In this example, we do experiments on the KQA Pro validation set as the golden answers are available. PE denotes how much percent of the golden answers are generated from the parsing and execution stages, which is the result of the parse-execute paradigm. After the two stages, we can obtain the execution context $C_i$ (see Eq. 2), which is the intermediate step-by-step reasoning process. We also report the recall of $[q_i, C_i]$, which represents what percent of the golden answers are contained in the questions and execution contexts. Please note that when we calculate the recall of $[q_i, C_i]$, we also include the Verify question instances since the model can generate "yes/no" answer only based on the question and context, and the "yes/no" is not necessary to appear in the context. From Table 2, we can observe that the percent of the golden answers in the questions and contexts

| Model | Overall | Multi-hop | Qualifier | Compari-son | Logical | Count | Verify | Zero-shot |
|---|---|---|---|---|---|---|---|---|
| Parsing | 90.72 | 89.69 | 84.83 | 95.65 | 89.60 | 88.41 | 93.51 | 89.21 |
| +Exe | 92.66 | 91.54 | 86.78 | 96.82 | 91.49 | **89.01** | 94.41 | **92.20** |
| +Exe+Ref | **93.82** | **92.93** | **89.83** | **97.57** | **92.99** | **89.01** | **95.30** | 91.69 |

Table 3: Ablation study of three stages on KQA Pro test set. +Exe denotes that we combine Parsing stage and the Execution stage. +Exe+Ref represents our full model with three stages.

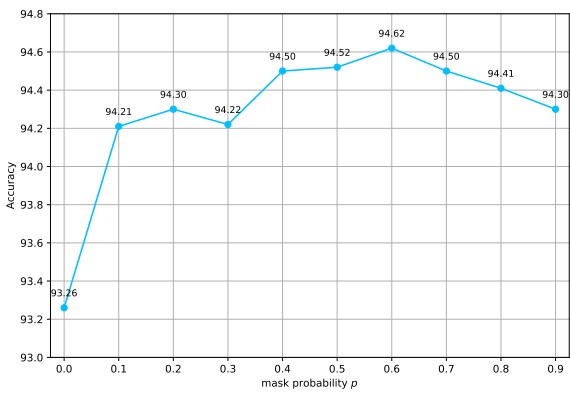

Figure 2: Ablation study of mask strategy on KQA Pro validation set.

is higher than those generated from the parsing and execution stages. It indicates that some of the golden answers in the intermediate results are not sufficiently utilized in the parse-execute paradigm. There is room for refinement stage to improve the performance only based on the questions and the proposed execution context. In summary, the refinement stage can further improve the ability of complex reasoning by fully exploring the intermediate step-by-step reasoning processes.

## 4.2 Ablation Study

**Three Stages:** In this set of experiments, we explore the performance of three stages respectively. Parsing denotes that we reproduce the BART KoPL from KQA Pro (Cao et al., 2022a) as our first stage.

Table 3 shows the comparison results. It can be observed that the execution stage is effective. During this stage, we can access the knowledge base and explicitly reason over the structure of the KB, which is significant to promote the generalization ability in the Zero-shot setting, the multi-hop capability and other fine-grained reasoning abilities. And we can further make improvement after we take advantage of the execution context to generate answer in the refinement stage.

**Mask Strategy:** In this set of experiments, we aim to explore why we leverage mask strategy in

the refinement stage. Figure 2 shows that if we do not mask the last returned answer during training, the answer generation model is likely to predict the answer based on the returned answer in the execution context and hardly obtain the performance gain. This is because the model will learn the spurious correlation between the last returned result and the golden answer.

To choose the mask ratio, we do experiments on the validation set using different values (i.e., $0.1, 0.2, \cdots, 0.9$). The accuracy increases as the mask probability $p$ of the returned result increases until $p$ reaches 0.6, and performance starts to drop after that. We can also see that the mask ratio is not sensitive. For example, the best performance is 94.62% ($p = 0.6$) and the worst performance is 94.21% ($p = 0.1$). The two results are close. Intuitively, in the situation where the information of the answer is almost provided, the model is less likely to attend to the question and the reasoning path in execution context, and hence lacks the generalization ability. Under the circumstance where the answer basically do not appear in the context, the model can still generate correct answers based on the reasoning path. Thus we select a reasonable value and set the mask probability as 0.6 to conduct our experiments on KQA Pro test set.

## 4.3 Case Study

To further understand our proposed approach PER-KBQA, we do case study to demonstrate how the three stages work. Shown in Figure 3, the predicted KoPL and execution contexts are organized in a structured text form to express the reasoning processes. Our proposed method can generate correct answer in the situation where the answer is contained in the execution context (the first case in Figure 3), and can predict the correct answer in the situation where the reasoning path is incomplete and the answer do not appear in the question or the execution context (the second case in Figure 3).

In the first case, the predicted KoPL, which is

Q: Who is shorter, David Brown (the point guard) or Stephen King (the one that was born in 86)?
Gold KoPL: Find(point guard) <func> Relate(position played on team / speciality <arg> backward) <func> Find(David Brown) <func> And() <func> Find(Stephen King) <func> FilterDate(date of birth <arg> 1986-03-06 <arg> =) <func> SelectBetween(height <arg> less)
Pred KoPL: Find(point guard) <func> Relate(position played on team / speciality <arg> backward) <func> Find(David Brown) <func> And() <func> Find(Stephen King) <func> FilterYear(date of birth <arg> 86 <arg> =) <func> SelectBetween(height <arg> less)
Parse pred ans: null
Parse+Exe pred ans : Stephen King
Execution Context: Find <arg> point guard <func> Relate <arg> position played on team / speciality <arg> backward <return> Robert Smith | Jason Kidd | Chris Paul | Jack Thompson | Chris Paul <func> Find <arg> David Brown <func> And <return> David Brown <func> Find <arg> Stephen King <func> FilterYear <arg> date of birth <arg> 86 <arg> = <return> Stephen King <func> SelectBetween <arg> height <arg> less <return> Stephen King
Parse+Exe+Ref pred ans: Stephen King
Gold ans: Stephen King

Q: What is the ending date that small forward is the position of Michael Jordan whose nickname is His Airness?
Gold KoPL: Find(Michael Jordan) <func> FilterStr(nickname <arg> His Airness) <func> Find(small forward) <func> QueryRelationQualifier(position played on team / speciality <arg> end time)
Pred KoPL: Find(small forward) <func> Find(Michael Jordan) <func> FilterStr(nickname <arg> Her Airness) <func> QueryRelationQualifier(position played on team / speciality <arg> end time)
Parse pred ans: None
Parse+Exe pred ans: None
Execution context: Find <arg> small forward <func> Find <arg> Michael Jordan <func> FilterStr <arg> nickname <arg> Her Airness <return> Michael Jordan <func> QueryRelationQualifier <arg> position played on team / speciality <arg> end time
Parse+Exe+Ref pred ans: 2003
Gold ans: 2003

Figure 3: Case analysis about our proposed method. We mark the error generations from the first stage and their ground truths in red. And the intermediate results are marked in green.

generated by the first parsing stage only conditioned on the given question, deviates slightly from the golden KoPL because of the lacking the specific information of the knowledge base. And the predicted KoPL can not be executed against the knowledge base to obtain the right answer. However, with the execution stage, we simply align the relations or attributes in the knowledge base to make the logical forms executable and can get the right answer. The refinement stage further utilizes the execution context to generate correct answer since the right answer is contained in the execution context.

In the second case, the generated KoPL from the first stage failed to give the right answer corresponding to the knowledge base. Even though we apply the alignment operation of the second stage, it can still not give the right answer because of the disturbance from other relations or attributes in the knowledge base. So the information of the correct answer is not contained in the execution context. However, the refinement stage can generate the correct answer based on the question and the reasoning path of the execution context. It indicates that utilizing the mask strategy to train the refinement model is beneficial for the model to store some specific information about the knowledge base. Thus the refinement model can generate answer correctly by attending to the question and the incomplete reasoning path of the execution context.

## 5 Conclusion

In this work, we propose a novel approach called PER-KBQA for complex question answering over the knowledge base. Our method consists of three stages: parsing, execution and refinement. The parsing stage generates logical forms conditioned on the question. The execution stage alleviates the problem of slight generation deviation via the alignment to the knowledge base and obtains the intermediate results. The refinement stage combines the logical forms and corresponding intermediate results as the reasoning process and regards it as execution context to generate the answer. Experiments on KQA Pro benchmark show that our method can significantly improve the performance in terms of most evaluated measures.

## 6 Limitations and Future Work

Our proposed method is skillful at answering complex questions over knowledge base, but our work also has limitations. Our approach relies on the logical forms that can provide transparent intermediate results and only the KoPL is transparent to the best of our knowledge. However, Cao et al. (2022b) had shown the promising results to transfer the KoPL to other complex KBQA benchmarks such as WebQSP (Yih et al., 2016), CWQ (Talmor and Berant, 2018). So in the future, we plan to adapt our method to other complex KBQA benchmarks.

## Acknowledgment

This work is supported by the National Natural Science Foundation of China (U1911203, U2001211, U22B2060), Guangdong Basic and Applied Basic Research Foundation (2019B1515130001, 2021A1515012172), Key-Area Research and Development Program of Guangdong Province (2020B0101100001).

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
