# OpenReview forum: "From Parse-Execute to Parse-Execute-Refine: Improving Semantic Parser for Complex Question Answering over Knowledge Base"
_EMNLP/2023/Conference — EMNLP 2023 Main_

### Official Review · Reviewer_poiZ · 2023-08-01

**Soundness:** 5

**Excitement:**

4: Strong: This paper deepens the understanding of some phenomenon or lowers the barriers to an existing research direction.

**Paper Topic And Main Contributions:**

The paper presents sota results on KQA Pro Knowledge Base Question Answering dataset. It relies on existing work that presented a parse-execute architecture but improves it by adding a refinement step. Multiple architecture variations are tested in an ablation study, clearly indicating that the refinement step yields performance improvements.

**Questions For The Authors:**

A) Does the Jaccard Similarity in 3.2 refer to a character-based similarity?

B) If I understood correctly in line 326: "spurious correlation" is not a spurious correlation as it is the best answer the initial model can come up with right? Maybe find another wording here. Same in 514

C) How is the accuracy for the Parsing only setup measured? You write that the information in the knowledge base is not used but the KoPL queries only produce an actual answer upon execution, right? My guess is that the <return> token and a subsequent answer is also generated alongside the query but this is not very clear and not shown in any of your examples. Please make it clear that this stage only uses the model weights when discussing the evaluation.

D) Does the prior work on KQA Pro not use any alignment? This may be answered somewhere in the paper but I can't seem to find it now.

**Reasons To Accept:**

The paper presents a clear improvement over existing work using the same benchmark datasets. It makes use of a novel parse-execute-refine architecture and shows clear improvements over existing work without introducing extreme complexity. While the work is currently limited to one dataset, the authors state that it can (with the use of existing methodologies) be applied and tested on other datasets.

The paper is written in a clear style with the main argument being very easy to follow. While there are some minor typos/grammar errors they did not impact my understanding.

**Reasons To Reject:**

The evaluation is (by necessity due to the focus on KoPL) limited to a single dataset and thus not comparable to a wide body of knowledge base QA work. This limitation, however, is clearly stated!

**Reproducibility:**

4: Could mostly reproduce the results, but there may be some variation because of sample variance or minor variations in their interpretation of the protocol or method.

**Reviewer Confidence:**

4: Quite sure. I tried to check the important points carefully. It's unlikely, though conceivable, that I missed something that should affect my ratings.

**Typos Grammar Style And Presentation Improvements:**

"The SPARQL 051
transduces natural language utterances into graph 052
query representations." this sentence does not make sense to me, the query language doesn't interact with natural language utterances, right?

101: "Observed from that" -> Based on those observations?

171 and 172: add space after TF-IDF and BM25 before citation

238 and 416: Bart -> BART

Table 3: Count column, two numbers are the same 89.01 -> should both be bold unless this is some copy-paste error

431: you claim significance, implying statistical significance which is not analyzed. Perhaps go with a more neutral word like "clear" instead. There are a couple more instances of this.

616: Kqa -> KQA

472: "how much percent of" -> what percentage of

---

> ### Author Rebuttal · Authors · 2023-08-28
>
> Thanks for your careful review and appreciating our work.
>
> To Question A: Yes, the Jaccard Similarity refers to character-based similarity.
>
> To Question B: According to your suggestion, we will remove the word and rewrite the sentences.
>
> To Question C: The Parsing only setup means that we only used the question-logical form pairs to train the semantic parse model and during inference time we can obtain the logical form given a question and execute it against the KB to get the final answer. The information about the knowledge base is not used during training the semantic parse model. Since the second stage may produce empty answer such as None, thus, in our experiments, we did not append the <return> token and a subsequent answer when there is an empty answer from the second stage.
>
> To Question D: To our best knowledge, we firstly apply alignment on KQA Pro.

---

### Official Review · Reviewer_yo9y · 2023-08-02

**Typos Grammar Style And Presentation Improvements:** NA
**Soundness:** 4

**Excitement:**

4: Strong: This paper deepens the understanding of some phenomenon or lowers the barriers to an existing research direction.

**Missing References:**

NA

**Paper Topic And Main Contributions:**

This paper aims to improve the performance of a semantic parser called KoPL, which has been used for complex knowledge base question answering (KBQA). Specifically, instead of only showing the final results to the final QA model, this paper argues that showing the middle steps could help the final QA model achieve overall performance.

**Questions For The Authors:**

1. Can you introduce more details about the alignment discussed in line 257?

**Reasons To Accept:**

1. Even though the proposed method is simple, it is sound and light-weight for this specific task (i.e., complex question answering over knowledge base)
2. The experimental design is thorough, encompassing a broad range of scenarios to validate the efficacy of the approach.

**Reasons To Reject:**

1. The impact of the proposed method is limited. The importance of the middle step for reasoning tasks is well known (e.g., CoT). This paper mainly focuses on extending the idea of using the middle reasoning step to the complex KBQA problem.
2. The overall technical paradigm of the proposed framework is slightly outdated. It would significantly strengthen the paper if the authors could include a discussion about using LLM language models on this task. (e.g., for parsing and the final merging)

**Reproducibility:**

5: Could easily reproduce the results.

**Reviewer Confidence:**

3: Pretty sure, but there's a chance I missed something. Although I have a good feel for this area in general, I did not carefully check the paper's details, e.g., the math, experimental design, or novelty.

---

> ### Author Rebuttal · Authors · 2023-08-28
>
> Thanks for your careful review and kind suggestions. Firstly, LLMs needs to rely on large scale of data and large model, while we used small language model that can make us easily train the sematic parser and the answer generator. Secondly, both CoT and our method leveraged the intermediate reasoning steps, but CoT needs to be used by LLMs to work. Instead, to achieve better reason capability, our method that used small language model needs more detailed design such as alignment and refinement. Thirdly, though LLMs is making huge success, we do not think that not leveraging LLMs is outdated.
>
> To Question 1: The detailed alignment step was described in Line 262 ~Line 172. The goal of the alignment process is to help execute the generated KoPLs successfully, which is to deal with the cases that the generated arguments do not appear in the knowledge base. Thus, we define functions to find the closest concepts, entities or relations in the knowledge base to replace those out-of-KB arguments that are generated. We use Jaccard Similarity to measure the distance for simplicity.

---

### Official Review · Reviewer_MbeL · 2023-08-11

**Soundness:** 4

**Excitement:**

3: Ambivalent: It has merits (e.g., it reports state-of-the-art results, the idea is nice), but there are key weaknesses (e.g., it describes incremental work), and it can significantly benefit from another round of revision. However, I won't object to accepting it if my co-reviewers champion it.

**Missing References:**

See "Reasons To Reject"

**Paper Topic And Main Contributions:**

This paper aims to enhance the semantic parser for complex KBQA by introducing a PER (parse-execute-refine) paradigm. This approach involves three stages: parsing, execution, and refinement. The parsing stage trains a BART neural parser to generate KoPL logical forms which are claimed to be transparent and explainable. The execution stage aligns and executes the logical forms over a knowledge base to obtain intermediate results of the sub-queries. In the refinement stage, these intermediate results from the second stage are demonstrated to another BART model together with the execution result of the complete logical form generated at the first stage, to achieve better end-to-end KBQA accuracy. The key contribution of the paper is the performance improvements of its proposed PER-KBQA approach, which can be credited to (a) the explicit alignment of the KoPL arguments to the knowledge base schema (i.e., relations and attributes), and (b) the answer refinement based on both the original execution results and the additional results from the intermediate logical form reasoning process. The results of the experiment indicate a significant improvement on the benchmark KQA Pro. Additionally, the results of the ablation study and case study provide supportive evidence for the paper's motivation.

**Questions For The Authors:**

1. Can you provide any insights or explanations as to why the zero-shot performance decreases after refinement?
2. Has your approach been tested on the non-IID data splits of the KQA Pro benchmark to validate compositional generalization?
3. Were the two cases presented in Section 4.3 selected randomly or based on certain criteria?
4. What is the average length of the execution context that is fed into the refinement model? Would the performance be enhanced by providing more than five results to the refinement model?
5. Is it possible that an LLM could outperform a fine-tuned BART as the refinement model due to its ability to incorporate both open-domain knowledge and KB for answer refinement?
6. I'm not certain if the previous works on KQA Pro utilized KB schema or ontology for alignment. If not, I believe the method's additional dependence on KB schema information should be addressed in the discussion.

**Reasons To Accept:**

The deviation problem commonly exists in today's neural semantic parsers. This paper proposes a simple yet effective approach PER-KBQA for alleviating this problem by involving the intermediate execution results to refine the errors in the generated logical forms, in order to improve the end-to-end accuracy for complex KBQA tasks. The motivation and methodology of PER is therefore quite easy to understand. Based on the experimental results, the performance gain also looks quite promising. The outcome of the ablation study & case study also sounds reasonable and can provide solid evidence to support the effectiveness of the proposed approach.

**Reasons To Reject:**

The performance improvements of PER-KBQA mainly come from two sides: the KB schema alignment at the second stage, and the answer refinement at the third stage using the intermediate results. Both ideas are not novel and have been explored in many previous KBQA works. For example, both (Herzig and Berant, 2018) and (Wang et al., 2021) suggest schema linking alongside semantic parsing to reduce the mismatch between the generated logical form structures and KB schemas. Using the intermediate reasoning results as supplementary supervision has also been proposed by previous works like (He et al., 2021) and (Cao et al., 2022).

Also, if the intermediate results are presented to the refinement model, the end-to-end accuracy may improve only because the model is memorizing more KB schema & ontology, which may lead to worse generalization in non-IID settings. As shown in the ablation study, the refinement stage actually impairs the zero-shot performance.

Ref:
Herzig, Jonathan, and Jonathan Berant. "Decoupling Structure and Lexicon for Zero-Shot Semantic Parsing." In Proceedings of the 2018 Conference on Empirical Methods in Natural Language Processing, pp. 1619-1629. 2018.
Wang, Bailin, Richard Shin, Xiaodong Liu, Oleksandr Polozov, and Matthew Richardson. "RAT-SQL: Relation-Aware Schema Encoding and Linking for Text-to-SQL Parsers." In Proceedings of the 58th Annual Meeting of the Association for Computational Linguistics, pp. 7567-7578. 2020.
He, Gaole, Yunshi Lan, Jing Jiang, Wayne Xin Zhao, and Ji-Rong Wen. "Improving multi-hop knowledge base question answering by learning intermediate supervision signals." In Proceedings of the 14th ACM international conference on web search and data mining, pp. 553-561. 2021.
Cao, Shulin, Jiaxin Shi, Zijun Yao, Xin Lv, Jifan Yu, Lei Hou, Juanzi Li, Zhiyuan Liu, and Jinghui Xiao. "Program Transfer for Answering Complex Questions over Knowledge Bases." In Proceedings of the 60th Annual Meeting of the Association for Computational Linguistics (Volume 1: Long Papers), pp. 8128-8140. 2022.


**Reproducibility:**

4: Could mostly reproduce the results, but there may be some variation because of sample variance or minor variations in their interpretation of the protocol or method.

**Reviewer Confidence:**

5: Positive that my evaluation is correct. I read the paper very carefully and I am very familiar with related work.

**Typos Grammar Style And Presentation Improvements:**

a. "KQAPro" at line 72, 73, 082, 121, 239 should be "KQA Pro".
b. be consistent about the wording of "pre-trained" or "pretrained" model
c. line 309: "Quesion" should be "Question"

---

> ### Author Rebuttal · Authors · 2023-08-28
>
> Thanks for your careful review and kind suggestions. Firstly, previous works have studied the alignment between generated logical forms and KB schemas, but almost of them needed to train a complex neural model to calculate the semantic similarity for alignment. However, in our situation, we performed the alignment operation during the execution process. Since the candidate pool is always large, we used the Jaccard similarity which is more efficient (Line 273-282). Secondly, the intermediate reasoning results were used as supplementary supervision in previous works, but in our approach, we utilized the intermediate results as context for further refinement instead of supervision information. Thirdly, our method did not perform worse generalization in non-IID settings. Instead, our method can make significant improvement on compositional generalization (see To Question 2). We will cite and discuss these references.
>
> To Question 1: This is because our method focuses on the overall metrics (e.g. Multi-hop, Qualifier, Verify), not just zero-shot. Hence, we did not design complex methods to improve the zero-shot performance. In this paper, to improve the generalization, we only mask the returned answer from Execution stage in a certain probability during training the refinement model. We can expect that the use of more complect and mature methods will improve generalization. We will explore this in our future work.
>
> To Question 2: Thanks for your constructive suggestions. Following KQA Pro(Cao et al., 2022a ), we conduct the experiments, which focus on compositional generalization to longer sequences or to greater compositional depths than have been seen in training. Since the train/val/test set of compositional generalization are not available, we re-split the train and val sets. We take the instances with short programs as training examples (length <= 9), and those with long programs as valid examples (length > 9). The experimental results are shown as below.
>
> | |BART KoPL   |   PER-KBQA(ours)|
> |--------|--------|---------|
> |Compositional Generalization|76.62 | **93.95**|
>
> The experimental results showed that our approach can generalize to non-IID data splits of the KQA Pro benchmark and can significantly improve the performance of compositional generalization.
>
> To Question 3: We randomly selected cases in Section 4.3
>
> To Question 4: The average length of the execution context is about 400. We conducted the experiments on the validation set to study whether the performance be enhanced by providing more than five results to the refinement model.
>
> |Number of intermediate results |  3 | 5 | 7|
> |---------|--------|--------|--------|
> |Val Acc | 94.46  |  **94.62**  |  94.20 |
>
> The experiments showed that much more intermediate results cannot improve the performance. An interpretation is that much more intermediate results may induce more noise for refinement and less intermediate results may decrease the recall of the gold answers, both of which may do harm to the performance.
>
> To Question 5: It is possible. However, directly applying the question and execution context as input for LLMs may not make improvement in the case that LLMs have not seen the knowledge base. Hence, other techniques such as in context learning and retrieval augmentation need to be further explored.
>
> To Question 6: We utilized the similarity score function to align with the KB. It is general and is not dependent on KB schema information.

---

### Meta-Review · Area_Chair_HMcS · 2023-09-26

**Recommendation:** 4

**Metareview:**

This paper presents a new parse-execute-refine framework that interleaves logical form generation with KB execution to let a LM take the execution context into account when generating an answer.

This method yields significant gains over comparable baselines on overall and most partial metrics, including significant gains across the board over the relevant LLM baseline. These are all evaluated on a single dataset.

While the results are promising, the paper could benefit from a more thorough comparison to related work that focused on similar tasks and datasets. Alignment and intermediate execution are not novel (they even preceed neural semantic parsing) and, while the approach here is novel, the paper would benefit from a deeper discussion of how it relates to other work.

---

### Decision · Program_Chairs · 2023-10-07

**Decision:**

Accept-Main

**Comment:**

This paper presents a new parse-execute-refine framework that interleaves logical form generation with KB execution to let a LM take the execution context into account when generating an answer.

This method yields significant gains over comparable baselines on overall and most partial metrics, including significant gains across the board over the relevant LLM baseline. These are all evaluated on a single dataset.

While the results are promising, the paper could benefit from a more thorough comparison to related work that focused on similar tasks and datasets. Alignment and intermediate execution are not novel (they even preceed neural semantic parsing) and, while the approach here is novel, the paper would benefit from a deeper discussion of how it relates to other work.